# Determinants of profiles of competence development in mathematics and reading in upper secondary education in Germany

**Micha-Josia Freund** [ORCID]*, **Ilka Wolter**◉, **Kathrin Lockl**◉, **Timo Gnambs** [ORCID]

Leibniz Institute for Educational Trajectories, Bamberg, Germany

◉ These authors contributed equally to this work.
* Micha.Freund@lifbi.de

## Abstract

The registered report was targeted at identifying latent profiles of competence development in reading and mathematics among $N = 15,012$ German students in upper secondary education sampled in a multi-stage stratified cluster design across German schools. These students were initially assessed in grade 9 and provided competence assessments on three measurement occasions across six years using tests especially developed for the German National Educational Panel Study (NEPS). Using Latent Growth Mixture Models, Using Latent Growth Mixture Models, we aimed at identifying multiple profiles of competence development. Specifically, we expected to find at least one generalized (i.e., reading and mathematical competence develop similarly) and two specialized profiles (i.e., one of the domains develops faster) of competence development and that these profiles are explained by the specialization of interest and of vocational education of students. Contrary to our expectations, we did not find multiple latent profiles of competence development. The model describing our data best was a single-group latent growth model confirming a competence development profile, which can be described as specializing in mathematical competences, indicating a higher increase in mathematical competences as compared to reading competences in upper secondary school. Since only one latent profile was identified, potential predictors (specialization of vocational education and interest) for different profiles of competence development were not examined.

## 1 Introduction

The introduction with the theoretical background and the method description are reproduced *verbatim* from the Registered Report Protocol [1]. All modifications to these sections are listed in the online supplement.

Language and mathematical competences significantly impact academic and professional success. Basic language competences (including reading competence) are at the core of learning and communicating [2], while basic mathematical competence (or mathematical literacy) is defined by the Organization for Economic Co-Operation and Development (OECD) ([3],

**Data Availability Statement:** As we work with secondary data, this dataset cannot be shared. However, the data is accessible to scientists fulfilling all necessary requirements of the national education panel study (NEPS) once they have concluded a Data Use Agreement with the Leibniz Institute for Educational Trajectories (https://www.neps-data.de/Data-Center/Data-Access). Once a scientist has access to the data, the data will be available on the NEPS-homepage (https://www.neps-data.de/Mainpage).

**Funding:** The publication of the preceding Registered Report Protocol was funded by the Open Access Fund of the Leibniz Association.

**Competing interests:** The authors have declared that no competing interests exist.

p.15) as the ability "to make well-founded judgements and to use and engage with mathematics in ways that meet the needs of that individual's life as a constructive, concerned and reflective citizen". Both competence domains are basic skills necessary for everyday life, which is why both reading and mathematical competences are often analyzed in educational research.

Students in secondary education display consistent development in reading and mathematical competences with a reduced growth rate towards the end of compulsory education [4–6]. The two domains are highly correlated in cross-sectional data in both lower [4] and upper secondary education [7]. Previous research on the relationship between the development in reading and mathematical competences demonstrated substantial correlations between the change trajectories in both domains throughout secondary education [4, 8]. However, at the end of mandatory education (i.e., in the years following Grade 9 in Germany), research on domain-specific competence development and especially on the relationship between the two domains through a longitudinal perspective is scarce.

Against this background, this paper aims to analyze the longitudinal trajectories of mathematics and reading competence by identifying profiles of competence development of students in Germany at the beginning of upper secondary education, commencing in Grade 9 until age 21/22. We expect these profiles to be either generalized profiles of competence development (i.e., similar development in both reading and mathematical competence) or specialized profiles of competence development (i.e., a higher development in either domain). In a previous study with students at the beginning of lower secondary school in Germany (Grades five to nine), we were unable to confirm specialized profiles of competence development in those domains [9]. However, based on the manifold options the German educational system offers in upper secondary school, a higher level of specialization is expected in this period of schooling. If the expected profiles of competence development are found, potential predictors of profile-membership are also analyzed.

## 1.1 Individual's characteristics as determinants of competence development in reading and mathematics

Certain student characteristics can influence the development of reading and mathematical competence development of all students. Some of these explain the high correlation between mathematical and reading competences. In this context, research has shown that underlying abilities such as working memory [10–12] and reasoning ability [13] impact both domains. For example, several studies discovered working memory to be substantially correlated to both language and mathematical competences [14–16]. In a recent meta-analysis by Peng et al. [17], working memory and reasoning abilities together accounted for over 50% of the variance in the relation between language and mathematics. Additionally, the correlation between mathematical and reading competences can be traced back to the fact that general language and reading competences are important for learning in general but also for acquiring mathematical knowledge and solving mathematical problems [2, 17–19].

Previous research has additionally shown that socio-demographic characteristics of the students impact their competence development. Mathematical and reading competences are highly correlated to the socio-economic status of students' parents even before elementary education [20] and throughout secondary education [21]. As a summary of studies by Shin and colleagues [4] shows, the gap between students from high and low socioeconomic backgrounds was displayed as an increase, a decrease, or a stagnation depending on the model, tests, and sample that were used. Hence, analyzing specific longitudinal effects of social background on profiles of competence development are difficult to work out. Nonetheless, the

socio-economic background can be seen as a determinant of both competence domains simultaneously, further indicating generalized profiles of competence development.

Moreover, differences in reading and mathematical abilities were confirmed for male and female students. Cross-sectional studies in this field depict that, on average, boys have higher mathematical and lower reading competence in Grade 9 compared to girls [22]. These interindividual (between-student) differences imply intra-individual (within-student) differences between the domains at least cross-sectionally. The pattern of the development of gender differences from a longitudinal perspective is less clear, with studies showing that gender differences decrease [23] or stagnate [24] in secondary education. Thus, while the effect of gender on cross-sectional competence differences seems quite clear, longitudinal effects are difficult to predict.

Socio-demographic characteristics are not the only individual determinants of competence development implying potential specialization. Affective-motivational (e.g., motivation [25, 26], interest [27]) or socio-cognitive (e.g., self-concept [27]) factors, which substantially vary between the domains [27, 28] and are related to the frequency of school-related or leisure time activities [29], also have impacts on competence development in mathematics and reading. For example, Ehrtmann, Wolter, and Hannover [28] showed that many sixth-grade students' interest in German and mathematics (as well as further vocational interest domains) can be classified as generalized high or low, but some students are located in a profile with high interest in mathematics and low interest in German, or a profile with high interest in German and low interest in mathematics. As aforementioned, due to the correlation of interest and frequency of activity in a domain, we expected that students more likely belong to a profile of specialized competence development if they are distinctively more interested in one of the domains than the other. The existence of both generalized and specialized profiles of interest overall implies the existence of these profiles in competence development as well.

## 1.2 Context characteristics as determinants of competence development in reading and mathematics

Finally, the learning context also plays a role in competence trajectories. That is, competence development in both domains is affected by the characteristics of teaching in the classroom and the type of school a student attends [30] but also by students' choices during their educational career. Variations in the development of mathematical and reading competences in upper secondary school might be enforced by specific characteristics of the German educational system. With the end of lower secondary school and compulsory schooling after the ninth grade, the German system offers multiple pathways in either further general education towards a university entrance certificate or vocational training and associated exams [31].

The German school system is best described as a highly tracked school system [32]. Starting mostly in Grade 5 with entering lower secondary education all school types (mainly: Hauptschule, Realschule, Mittelschule, Gesamtschule, Gymnasium) focus on providing their students with general education until the end of compulsory education after Grade 9. These school types, however, differ mainly in their overall level of curricula but similarly focus on mathematics and reading competence. Starting with upper secondary education after Grade 9 some students decide to aim for a university entrance certificate, whereas other students leave the general education system and enter vocational training or alternative paths. Additionally, even students staying in general education have more options to decide between basic and advanced courses [33], which also determines parts of their exams at the end of schooling.

Students in vocational training [34] are already selecting their occupations and should more likely show specialized competence development. Hence, their competence profiles are

expected to be specialized on either mathematics or reading competence throughout their vocational training due to the focus of their apprenticeships on job-specific skills. Similarly, after finishing upper secondary education with a university entrance certificate, students entering university can decide on a university course focusing on either predominantly language- or reading-related competences (e.g., arts or language studies) or mathematical competences (e.g., science, technology, engineering, or mathematics; i.e., STEM) [35]. We thus expected that students in specific vocational training or university study programs are more likely to be specialized in their competence development in reading and mathematics than students not in specific vocational training or university courses. Overall, the increased variety and larger number of choices on pathways and courses in upper secondary education further strengthens the argument that there are specialized profiles of competence development throughout the course of upper secondary education.

## 2. Hypotheses

Against this background, we expected to identify not only a generalized profile of competence development with a similar trajectory for mathematical and reading competence but also specialized profiles of competence development at the beginning of upper secondary education. More specifically, we expected two specialized profiles of competence development, which are differentiated into a predominantly mathematical competence and a predominantly reading competence profile.

*Hypothesis 1*: *There are one generalized and two specialized profiles of competence development.*

Learning environments of students after Grade 9 should have an impact on their likelihood of belonging to either specialized or generalized profiles of competence development. Specialized interest can be interpreted as a higher likelihood of investing leisure time to acquire either mathematical or reading competences which in turn leads to higher competences in the specific domain. Similarly, students might focus more on one domain through further education. Vocational education after Grade 9 and higher education after Grade 12 can prepare for a career in a specific work sector or job. Since that work sector or job might demand a higher competence level in either reading or mathematics, a high specificity of vocational or higher education could lead to a higher likelihood of ending up with a specialized profile of competence development.

*Hypothesis 2*: *Students with interests predominantly in one domain, reading or mathematics, are more likely to specialize in that domain than students with an unspecialized interest.*

*Hypothesis 3*: *Students who choose an occupation or a university program in a STEM field in school more likely belong to a specialized profile in mathematics than in reading. Corresponding to this, students who choose an occupation or a university program identified as reading-centered are more likely to belong to a specialized profile in reading than in mathematics.*

## 3. Materials and methods

### 3.1 Sample

The study used data from a sub-sample (starting cohort Grade 9) of the German *National Educational Panel Study* (NEPS [36]), which examined representative samples of students from secondary schools across their educational careers. The National Educational Panel Study is a study "collecting longitudinal data on educational processes and individual competence development across the entire life span from early childhood to late adulthood" [36] in Germany

across different age groups in multiple datasets. In NEPS, students were sampled in a multi-stage stratified cluster design [37]. They were examined via questionnaires and were tested with standardized competence tests [38]. Additionally, both educators and teachers were asked to answer questionnaires to contribute additional information. The present study ($N$ = 15,012), focused on students who were initially tested in mathematics and reading in grade 9 (age $M$ = 15.2 years, SD = 0.6) and, subsequently, received competence tests in mathematics and reading at three-year intervals. The sample included 49.8% female students and 21.2% students with a migration background. Finally, 36.3% of students attended the higher school tracks (i.e., Gymnasium or the equivalent branch of a comprehensive school).

## 3.2 Knowledge of data

The lead author had not previously worked with this dataset. All theories and hypotheses, as well as details on the methodological approach, were based on a thorough literature review and prior research on other samples of the NEPS, including a currently unpublished paper with a similar aim in a mutually exclusive dataset with students in lower secondary education. The co-authors had previously worked with the dataset, albeit on topics unrelated to the present research. All publications using NEPS data published by the authoring team can be found at https://www.neps-data.de/Project-Overview/Publications (filtering for starting cohort 4). Furthermore, the co-authors had also contributed to some unpublished papers, which used the present dataset. However, none of the authors conducted analyses pertaining to this pre-registration, including identifying profiles of competence development, or identifying profiles across multiple domains in upper secondary education. The authors thus had no knowledge of the results of this study prior to publishing this report. All information used in the protocol was derived from the documentation available online (https://www.neps-data.de/Data-Center/Data-and-Documentation/Starting-Cohort-Grade-9/Documentation).

## 3.3 Instruments

In ninth grade, mathematical and reading competences were measured in a class-context, whereas later assessments were conducted individually in the students' private homes by trained test supervisors. Information on students' backgrounds, as well as on predictor variables, was taken from a questionnaire answered by the students.

**3.3.1 Mathematical competence.** Mathematical competence tests with items from four content areas and six cognitive components were specifically developed for use in the NEPS [39]. The mathematical tests at the beginning of Grades 9, 12, and three years after Grade 12 consisted of 22, 21, and 21 items, respectively [40–42]. They included simple and complex multiple-choice items as well as short constructed responses. Item response theory was used for scaling the tests [43]. Weighted maximum likelihood estimates (WLE) [44] and linking across grades with the help of overlapping items were used to attain student proficiencies [45]. Reliabilities of the WLEs in the three grades were .81, .77, and .75, respectively. To compare the competences in the two domains, the WLEs were standardized according to the mean and standard deviation in Grade 9.

**3.3.2 Reading competence.** Reading competence tests in NEPS were constructed according to a theoretical framework with three cognitive requirements and five text types [46]. These tests were administered at the end of Grade 9, beginning of Grade 12, and three years after Grade 12. They consisted of 31, 28, and either 23 or 27 items, respectively. The number of items in the last test differed because of different difficulty-tiered booklets depending on prior reading competence levels [47–49]. The different tests were placed on a common scale using an anchor-test design [45] to allow for valid longitudinal change analyses. Reliabilities of the

WLEs for reading competence were .81, .80, and .77, respectively. The WLEs were standardized according to the mean and standard deviation in Grade 9.

**3.3.3 Additional variables.** To test hypotheses two and three, we included further variables in our analyses. To measure students' interest in academic domains (mathematics and German) in NEPS, a scale was adapted from Baumert and colleagues [50]. Students were asked four items per domain in Grade 9 on their interest in spending time on mathematics and literature. The four questions for each domain were then turned into a scale. After z-standardizing the scales, a difference score between the interests in the two domains was calculated and used as a metric scale to indicate specialization of interest.

Additionally, to analyze whether students spent significant time in reading or mathematically specialized education, all episodes of schooling, training, or studying that were at least six months long were considered. Each of these episodes was classified as either language specialized, mathematics specialized, or generalized (i.e., not specialized to either domain). Vocational trainings that were defined as STEM (science, technology, engineering, or mathematics) occupations by the Federal Employment Agency of Germany [51] and university programs in the fields of mathematics, natural sciences, and engineering [35] were coded as specialized in mathematics. Vocational trainings in the area of law, print-media, archives, and libraries as well as university programs in the fields of language and cultural studies, were coded as specialized in reading. All other episodes were coded as generalized (or unspecialized) episodes. Once every episode was coded, students were checked whether they spent significant time (at least six months) in only one of the two specialized areas (thus being specialized) or in both or in none (being generalized). This was ultimately combined in two separate dichotomous variables, each indicating one of the two specialization areas and both being mutually exclusive.

In addition to these predictor variables, several additional variables were necessary that were used for imputation in addition to competence and predictor variables. These variables included unique identifiers for the student and their school. Gender was already available in the dataset. The age of students was calculated in months by subtracting the month and year of the test in Grade 9 from their birth month and year. The highest occupational prestige of the parents (defined as a parent questioned in a questionnaire and their partner) using the International Socio-Economic Index (ISEI) of Occupational Status [52], and the highest number of years in education of the parents using the CASMIN (Comparative Analysis of Social Mobility in Industrial Nations) classification [53] were used as social background characteristics of students. To create a variable accounting for the type of school in Grade 9, all schools leading to university entrance qualification (i.e., Gymnasium, and the equivalent branch of comprehensive schools) were differentiated from all other types of schools.

Migration background was recoded to compare students with a first- or second-generation migration background (i.e., either students themselves or at least one parent born in another country) to all other students. A scale of interaction language in different contexts was created by taking the average of six variables on a students' interaction language: with their mother, with their father, with their siblings, with their best friend, at the schoolyard, and of the parents with each other. The domain-specific self-concept of students was also considered using variables adapted from Kunter and colleagues [54]. This questionnaire included 10 items on the self-concept of students in German and mathematics (five items each). Finally, a test on reasoning abilities [55] was included in the dataset. The original test included 12 items and examines if students can identify the right element to complete a given figural sequence. An overview of all variables can be found in Table 1.

**Table 1. List and description of all variables used in this study.**

| Variable | | Necessary transformation | Range of values |
|---|---|---|---|
| *Competence* | | | |
| Reading competence | Grade 9 | z-standardization | -∞ to +∞ |
| | Grade 12 | | |
| | Grade 12 + 3 years | | |
| Mathematical competence | Grade 9 | | |
| | Grade 12 | | |
| | Grade 12 + 3 years | | |
| *Predictors* | | | |
| Specialization of interest | | Creation of scale | -∞ to +∞ |
| Specialization of education | | Creation of scale | -1, 0, 1 |
| *Controls* | | | |
| Gender of the student | | - | 0, 1 |
| Migration background | | Dichotomization | 0, 1 |
| Type of school in grade 9 | | Dichotomization | 0, 1 |
| Highest CASMIN of parents | | Creation of scale | 9 to 16 |
| Highest ISEI of parents | | Creation of scale | 16 to 90 |
| Interaction language of students | | Creation of scale | 0 to 3 |
| *Additional auxiliary variables (for imputation)* | | | |
| Age of students at first testing | | Calculation | 0 to +∞ |
| Self-concept in German | | - | 1 to 4 |
| Self-concept in mathematics | | - | 1 to 4 |
| Reasoning ability of students | | - | 0 to 16 |

## 3.4. Statistical analyses

An overview over the planned statistical process, including the used datasets and variables at each step, can be found in Fig 1.

**3.4.1 Latent change analyses.** Longitudinal competence development was analyzed using linear latent growth models (LGM) [56]. The basic model provided information about the initial competence (intercept) and development (slope) of all students. Specifically, a dual-process LGM (with two slopes and two intercepts) was specified to acknowledge both mathematical and reading competences. This model was estimated in Mplus version 8 [57] using a maximum likelihood estimator with 4,000 initial stage starts and 1,000 final stage optimizations. The constraints for the slope parameters were zero, three and six years for the three waves respectively. Then, latent growth mixture modeling (LGMM) [58, 59] identified the different profiles of competence development. As the focus of this study was on the development of students (and not initial competence levels) our model only used the mean LGM slopes of mathematical and reading competences to allocate profiles of competence development. As such, the intercepts in both domains were constrained across all profiles.

**3.4.2 Dealing with missing values.** To account for the dropouts in the data of NEPS, we used a multiple imputation approach [60]. We imputed missing values 30 times using predictive mean matching in the Stata-package ICE [61]. For imputation, we used age, type of school in grade 9, interaction language of the students, migration background, reasoning abilities, the domain-specific self-concept in German and mathematics, the highest ISEI and the highest CASMIN of the parents in addition to the competence tests in mathematics and reading for each grade and the aforementioned predictor variables (gender, specialization of further educational paths, and specialization of interest in mathematics or reading).

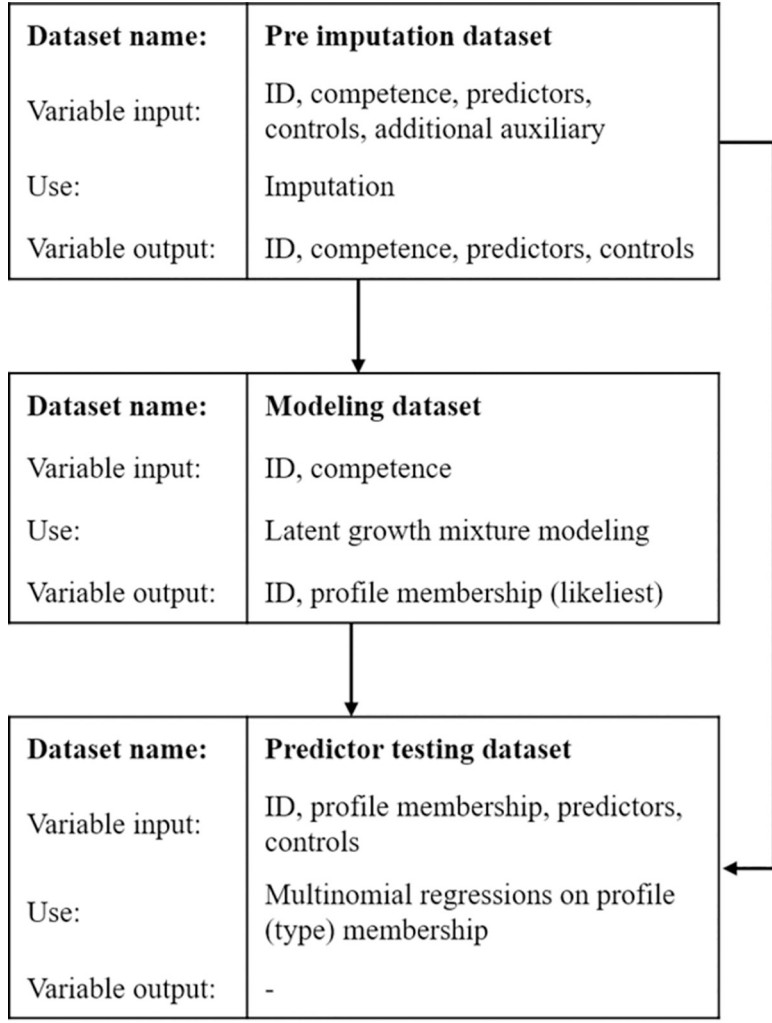

**Fig 1. The three statistical steps, necessary datasets and variables.**

**3.4.3 Model selection.** To identify the optimal number of profiles, we fit different LGMMs with 1 to 10 classes. Then, we excluded models with profiles including less than 5% of the students. Smaller profiles are likely difficult to replicate and seem to have negligible practical relevance. In a next step, the model with the best fit was chosen using the Bayesian Information Criterion (BIC) [62] and the Lo-Mendel-Rubin Likelihood Test (LMRT) [63, 64]. The model with the lowest BIC and a significant LMRT can be interpreted as the model with the best fit. A significant ($\alpha$ = .05) LMRT indicates that a model with $k$ profiles provides a better fit than a model with $k$-1 profiles. All criteria for model selection are summarized in Table 2.

**Table 2. Criteria for model selection.**

| Name | Type of criterion | Decision making process |
|---|---|---|
| Profile size | Exclusion criterion | Profile size of every profile at least 5% |
| BIC | Fit index | Lowest BIC indicates best fit |
| LMRT | Fit index | Last significant LMRT indicates best fit |

**3.4.4 Interpretation of profiles.** The basic LGM acted as a baseline to interpret the profiles of the other models. We took the sum of both slopes in the LGM and divided it by 4. This resulted in a threshold of 0.045 which served as a criterion for interpretation. If the difference between the two slopes in a profile was greater than this criterion, students differed more in their development between the domains than the average student develops within half a year. Profiles with a higher difference were interpreted as specialized profiles of competence development while profiles with a lower difference were interpreted as generalized. All profiles fit into one of these three types of profiles, as only this difference between the slopes (and not the absolute level of slopes or intercepts) was relevant for profile interpretation.

However, it was possible, that this classification resulted in several profiles of the same type. For example, it was conceivable that two specialized profiles appear that simply differ in their degree of specialization (i.e., the amount of difference in slopes). However, differences within profile types were not the focus of the present study. Therefore, for the prediction analyses, if more than one profile of a type was identified, these profiles were then merged into a single profile type. For example, two generalized profiles, two profiles specialized in mathematics and one profile specialized in reading would be condensed into three profiles, each containing all original latent profiles of their type.

**3.4.5 Testing predictors.** If we identified both generalized and specialized profiles, we were able to test the influence of the predictors on the likelihood of belonging to each class via a three-step approach [65]. In this approach, the most likely latent class and the measurement errors for each student (calculated in step one in the LGMM) are saved as manifest variables (step two). The effect of the predictors on the likelihood of class-membership is then tested via multinomial regression (step three). In this regression, both predictors and several additional control-variables were used (see Table 1). As an inference criterion for the effect of interest and educational pathways, we used an a priori significance level of 1%.

## 3.5 Open practices

Details on the study material and the assessment procedure are available at https://neps-data. de. The analyzed data is owned by a third party and can thus not be accessed through direct means. However, it is freely available to scientists after signing a data use agreement and is provided at http://dx.doi.org/10.5157/NEPS:SC4:11.0.0. The computer code used to generate the reported results can be accessed at https://osf.io/x67bh/?view_only=77d0d99f497f43c3a2eb4 66f9b072553. The study was preregistered at [1].

## 4. Results

### 4.1 Descriptive analysis

In Table 3, the means, variances, and correlations of the imputed competence scores (z-standardized in reference to the mean and standard deviation in grade 9) are provided. Mathematics competence exhibited a substantially stronger increase across the six years ($d = 0.77$) as compared to reading competence ($d = 0.32$). However, in both domains, competences increased more strongly in Grades 9 and 12 (up to graduation from secondary education) as compared to the three years following Grade 12 (after leaving school). This might question the assumption of linear growth during the observational period. The standard deviations in mathematics changed very little in the six years following Grade 9, while the standard deviations in reading showed a slight but consistent decrease after Grade 9. Thus, individual differences in reading abilities decreased across the six years. Correlations between the two domains were moderately high across domains and measurement occasions.

**Table 3. Means, standard deviations, and correlations of competence tests.**

| | Grade | *M* | *SD* | Correlations | | | | | |
| | | | | Mathematics | | | Reading | | |
| | | | | 9 | 12 | 12+3 | 9 | 12 | 12+3 |
|---|---|---|---|---|---|---|---|---|---|
| **Mathematics** | 9 | 0.00 | 1.00 | | | | | | |
| | 12 | 0.56 | 0.92 | .70 | | | | | |
| | 12+3 | 0.77 | 1.00 | .68 | .68 | | | | |
| **Reading** | 9 | 0.00 | 1.00 | .54 | .44 | .47 | | | |
| | 12 | 0.21 | 0.81 | .56 | .50 | .53 | .65 | | |
| | 12+3 | 0.32 | 0.77 | .55 | .51 | .59 | .58 | .65 | |

*Note*. Competence scores were *z*-standardized within domain in reference to Grade 9. All correlations are significant at a 99.9% significance level.

## 4.2 Latent growth modeling

**4.2.1 Latent growth analysis.** The LGM showed average initial competence levels (intercepts) of 0.06 (*SD* = 0.82) in mathematics and 0.03 (*SD* = 0.78) in reading. Development of reading competences can be described with an average slope of 0.05 logits (*SD* = 0.05) per year, while the average development in mathematics was distinctively larger at 0.13 logits (*SD* = 0.03). Since the difference of the two slopes within the LGM was 0.08 and, thus, distinctively larger than our criterion for inferring specialization (0.045), the overall development of the students is interpreted as specializing in mathematics. The standard deviation of the two slopes was very small, which also implies that most students developed similarly with only small inter-individual differences.

The correlations between the latent growth factors (see Table 4) indicated a large relationship between both domains and their development across time. The two intercepts correlated at .83 and, thus, showed that reading and mathematics were cross-sectionally substantially associated. Additionally, both slopes were negatively related to the respective intercepts, -.27 and -.73 for mathematics and reading, respectively. This might indicate that especially in reading initial differences in competence reduced over time.

**4.2.2 Latent growth mixture model.** To identify the number of profiles, LGMM solutions with 1 to 5 profiles were compared (see Table 5). Both the LMRT and the BIC suggested a two-profile solution in which both profiles were specialized in mathematics. However, this solution did not meet our threshold of a minimum profile size (5%). The smaller (and more specialized) profile was estimated to include only 1.6% of the students. This led to only 70 students (0.5%) exhibiting this profile as their most likely profile. Due to the profile size criterion, a single profile solution, that is, the aforementioned LGM was preferred over any LGMM solution.

**Table 4. Means, standard deviation, and correlations of the latent intercepts and slopes.**

| | | *M* | *SD* | Correlation | | | |
| | | | | Mathematics | | Reading | |
| | | | | intercept | slope | intercept | slope |
|---|---|---|---|---|---|---|---|
| **Mathematics** | Intercept | 0.06 | 0.77 | | | | |
| | Slope | 0.13 | 0.02 | -.27 | | | |
| **Reading** | Intercept | 0.03 | 0.71 | .83 | -.33 | | |
| | Slope | 0.05 | 0.04 | -.40 | .65 | -.73 | |

*Note*: All correlations are significant at a 99.9% significance level.

**Table 5. Model fit and profile sizes of the LGM and LGMMs.**

| Groups | BIC | Entropy | average LMRT | Group size (based on estimated probabilities) | | | | |
|---|---|---|---|---|---|---|---|---|
| | | | | P 1 | P 2 | P 3 | P 4 | P 5 |
| 1 | 191,161 | 1.00 | - | **15,012** | | | | |
| 2 | **191,120** | 0.93 | **0,001** | 14,770 | 242 | | | |
| 3 | 191,123 | 0.92 | 0,075 | 14,616 | 232 | 164 | | |
| 4 | 191,145 | 0.89 | 0,174 | 14,308 | 323 | 195 | 187 | |
| 5 | 191,166 | 0.90 | 0,302 | 14,251 | 316 | 230 | 197 | 18 |

*Note*. Bold indicates best values for a criterion, underlining indicates values being below our set threshold for acceptable entropy ($> 0.7$) or profile size ($> 5\%$).

To examine whether these results were specific to our chosen modeling approach, we also estimated respective latent class growth analyses (LCGA). A summary of these results can be found in the online supplement. While these models exhibited profiles of useful size up to a 4-profile solution, the model fit of these models were distinctively worse than the model fit of the LGM and LGMMs. Thus, while profiles might be found using LCGAs, they were much worse at explaining the overall variance of competence development than the LGM, which simply allowed students to differ without specifying qualitatively distinct groups.

## 4.3 Prediction of latent growth

The second and third hypotheses outlined in the Registered Report Protocol [1] referred to predictors of students' most likely profile membership. However, as only one relevant profile was identified, these hypotheses cannot be addressed further.

## 5. Discussion

This present study aimed at identifying profiles of competence development in German upper secondary education and potential predictors for these profiles. The expectation of finding at least one generalized profile that is characterized by a similar competence development for reading and mathematics was based on high correlations between competence domains and their development identified in previous research [4, 7, 8]. Meanwhile, longitudinal specialized profiles were expected, among other reasons, due to specialization found in domain-specific interest [27, 28] and in vocational training [34] or areas of higher education–both factors that can be connected to competence development.

Similar to most studies analyzing the longitudinal relationship between competences in mathematics and reading [4, 8], the two domains were substantially correlated in this present study. With respect to the cross-sectional distributions, competences within and across waves were moderately to highly correlated to each other. Similarly, in the LGM competences were cross-sectionally highly correlated and also showed correlated change. These moderate to high correlations were expected since many factors (such as working memory [14–17], reasoning ability [17], and parental background [20, 21]) have an overlapping impact on reading and mathematical competences to a certain degree.

In the LGM, the two latent intercepts as well as the two latent slopes were correlated with each other. However, initial competence levels (i.e., intercepts) were negatively correlated to competence development (i.e., slope) both within and across domains. This result shows that students with initially higher competences improved slower on average. The negative relationship was especially pronounced within reading competences. This finding suggests that

towards the end of secondary education, students with higher competences cannot expand or keep up their competence advantage, especially in reading competences.

The LGMM did not provide a solution with more than one profile having a relevant size (at least 5% of the sample). Due to this, the LGM was chosen as the model with the best overall fit. Hypothesis 1 of finding multiple profiles of competence development was therefore rejected. Hypothesis 2 and hypothesis 3 were not tested, because the hypothesized multiple profiles were not identified. The confirmation of only one latent profile with specialized development in mathematics is also in line with the high correlations between the two competence domains. The findings also indicate that prior group differences (such as cross-sectional gender differences [22]), vocational specialization for some students, and specialization of affective-motivational determinants (e.g., interest [27, 28]) show a different (potentially indirect or long-term) impact on competence development than expected.

The results of our study confirm that the overall development in upper secondary school can be described as specialized in mathematics, with students developing faster in mathematics than in reading competences. The average development in mathematics every year is quite substantial, whereas reading competence develops to a slower degree at the end of secondary education. Even though the expected profiles of competence development were not confirmed in this study, there is a specialization towards mathematics throughout upper secondary education and this type of difference between the two domains can be partially attributed to a higher focus of vocational education on mathematics-related as compared to reading-related tasks. While only about 955 (6%) of the students in this study were found studying or training in reading specialized fields, specialized education in mathematics (i.e., STEM fields) was much more common with 4620 of the students (31%).

## 5.1 Implications and limitations

Future research on the development of competence profiles should focus on the specialization on mathematics in upper secondary, and tertiary education. Even though the findings of this study suggest a beneficial development in the area of mathematics for all students, minority students or women are still confronted with multiple challenges regarding their access to and persistence in STEM fields (e.g., [66]). There is an ongoing discussion and need for interventions regarding the access of women and minority students [67], especially since STEM fields are characterized by higher prestige and income (e.g., [68]). Therefore, the results lead to certain practical implications for researchers and educators interested in the cognitive development of adolescents. On one side, it is clear, that reading and mathematical competences are highly correlated. Hence, it can be argued that both reading and mathematical competences (or the understanding of complex texts and the understanding of complex mathematical problems) must be fostered to a similar degree. On the other side, the higher development in mathematical competence might point us in a different direction as students use their previously acquired competences in reading and mathematics to further improve in mathematics while reading competences might already reach a plateau throughout upper secondary education. This finding suggests a higher focus on promoting mathematical education to support students especially in this period of rapid improvement. Yet, students struggling with reading should be promoted regardless their improvements in mathematics. However, regarding any focus on mathematical or reading competence development, it is especially important to monitor the progress of all students. This includes not promoting gender stereotypes (as previously seen in interest-profiles [28] or in gender differences in competences [22]), helping students that start out with low competence in mathematics, and enabling all interested students to enter courses and fields that are focused on mathematics.

Also, future research might learn from a methodological point of view from this study. While multiple (relevantly sized) profiles cannot be found with the used data, different aspects such as different tests and more test repetitions can be seen as potential improvements for future studies. Using different approaches in test development might be favorable for this research question as the high correlation between reading and mathematical competences can partly be explained by the high *language* parts within the mathematical test. Many mathematical questions are presented as text-based questions and thus correct answers are partially conditional on reading competence. Moreover, additional measurement points for competence tests in both domains would allow for an analysis of exponential growth in addition to linear growth or additional linear growth factors that were modelled in the present study. Potentially, these additions would help to better describe the actual competence development of students than within the framework of one linear factor as used in this study.

Furthermore, the longitudinal development of cross-sectional competence profiles using a latent transition analysis might also be an interesting approach to analyze how initial competence levels and later competence levels are correlated within specific groups of students. Additionally, the effects of predictor variables might be analyzed with a different method (e.g., a linear regression on the growth parameters of the LGM) to analyze whether the variance can be explained by different factors and predictors. While the variance in specialization is not distinctive enough to identify profiles of competence development, the growth parameters can still be used as a dependent variable in linear regression analysis. Finally, such a prediction of the Latent Growth Model can also help explain the difference in growth between reading and mathematical competence by explaining which predictors affect one of the domains stronger than the other one.

Finally, the results of this paper might be seen with several drawbacks and limitations. The analyses might be hard to generalize due to the context of the German educational system. The German education system is tracked and one of the very few developed nations with vocational education as part of upper secondary education [32]. This vocational education system in Germany and the high proportion of mathematics-related work fields might be the reason for the overall specialized development in mathematics. Another limitation might be the high number of missing values in the sample. Due to the nature of a transitory educational phase (students leaving general education for vocational education), many students exited the study between the first and the second wave. We tried to keep the sample representative through multiple imputations using several additional auxiliary and control variables. However, the fact that students' characteristics changed severely before and after imputation indicates selective dropout. While this dropout is addressed through the imputation, it is always possible that the imputation model does not include all necessary variables and leaves out observed or unobserved variables that account for dropout.

## 6. Conclusion

Overall, our findings show that even though students differed in their initial competence and their competence development, the best way to explain the competence development of German upper secondary education students is to view them in an overall growth model. This model can be described as overall specialized in mathematics, yet, there is no indication of a generalized profile of similar development in mathematics and reading, or a profile of specialized development in reading. The results of this study differ from our previous research identifying one main generalized profile in lower secondary education [9] with a similar development in both domains. However, it adds to the previous finding, that even though there are intra-individual differences within students, they are not resulting in distinct profiles

of competence development. Future research should confirm the robustness of these findings in different contexts (i.e., educational systems) and samples.

## Supporting information

**S1 File. Online supplement: Contains all supporting tables.**
(DOCX)

## Acknowledgments

This paper uses data from the National Educational Panel Study (NEPS): Starting Cohort Grade 9, *doi:10.5157/NEPS:SC4:11.0.0*. From 2008 to 2013, NEPS data was collected as part of the Framework Program for the Promotion of Empirical Educational Research funded by the German Federal Ministry of Education and Research (BMBF). As of 2014, NEPS has been carried out by the Leibniz Institute for Educational Trajectories (LIfBi) at the University of Bamberg in cooperation with a nationwide network.

## Author Contributions

**Conceptualization:** Micha-Josia Freund, Ilka Wolter, Kathrin Lockl, Timo Gnambs.

**Data curation:** Micha-Josia Freund.

**Formal analysis:** Micha-Josia Freund.

**Writing – original draft:** Micha-Josia Freund.

**Writing – review & editing:** Micha-Josia Freund, Ilka Wolter, Kathrin Lockl, Timo Gnambs.

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
