## [Decision Letter · Decision Letter 0]

15 Jul 2021

PONE-D-21-17237

Profiles of competence development in upper secondary education and their predictors

PLOS ONE

Dear Dr. Freund,

Thank you for submitting your manuscript to PLOS ONE. After careful consideration, we feel that it has merit but does not fully meet PLOS ONE’s publication criteria as it currently stands. Therefore, we invite you to submit a revised version of the manuscript that addresses the points raised during the review process.

The three reviewers indicated that the paper is of high quality and adheres well to the outlined protocol. However, the reviewers also indicated some smaller issues with the manuscript that need to be addressed before the paper can be accepted for publication. Take a look at the provided comments and feedback, I hope you find them useful for improving your manuscript. I'll be looking forward to receiving the revision version.

We look forward to receiving your revised manuscript.

Kind regards,

Vitomir Kovanovic, Ph.D.

Academic Editor

PLOS ONE

Journal Requirements:

"From 2008 to 2013, NEPS data was collected as part of

 the Framework Program for the Promotion of Empirical Educational Research funded by the

German Federal Ministry of Education and Research (BMBF). As of 2014, NEPS has been

carried out by the Leibniz Institute for Educational Trajectories (LIfBi) at the University of

Bamberg in cooperation with a nationwide network."

"The publication of the preceding Registered Report Protocol was funded by the Open Access Fund of the Leibniz Association."

3. We noted in your submission details that a portion of your manuscript may have been presented or published elsewhere. [As mentioned before, the Introduction and Methodology has been published as part of the Registered Report Protocol. ] Please clarify whether this [conference proceeding or publication] was peer-reviewed and formally published. If this work was previously peer-reviewed and published, in the cover letter please provide the reason that this work does not constitute dual publication and should be included in the current manuscript.

Reviewers' comments:

Reviewer's Responses to Questions

**Comments to the Author**

1. Does the manuscript adhere to the experimental procedures and analyses described in the Registered Report Protocol?

If the manuscript reports any deviations from the planned experimental procedures and analyses, those must be reasonable and adequately justified.

Reviewer #1: Yes

Reviewer #2: Yes

Reviewer #3: Yes

2. If the manuscript reports exploratory analyses or experimental procedures not outlined in the original Registered Report Protocol, are these reasonable, justified and methodologically sound?

A Registered Report may include valid exploratory analyses not previously outlined in the Registered Report Protocol, as long as they are described as such.

Reviewer #1: Yes

Reviewer #2: Yes

Reviewer #3: No

3. Are the conclusions supported by the data and do they address the research question presented in the Registered Report Protocol?

The manuscript must describe a technically sound piece of scientific research with data that supports the conclusions. The conclusions must be drawn appropriately based on the research question(s) outlined in the Registered Report Protocol and on the data presented.

Reviewer #1: Yes

Reviewer #2: Partly

Reviewer #3: Yes

4. Have the authors made all data underlying the findings in their manuscript fully available?

Reviewer #1: Yes

Reviewer #2: Yes

Reviewer #3: Yes

5. Is the manuscript presented in an intelligible fashion and written in standard English?

Reviewer #1: Yes

Reviewer #2: Yes

Reviewer #3: Yes

6. Review Comments to the Author

Please use the space provided to explain your answers to the questions above. (Please upload your review as an attachment if it exceeds 20,000 characters)

Reviewer #1: Can you explain better what is the German National 171 Educational Panel Study (NEPS)? What kind of activities have students to answer?

Reviewer #2: This is an interesting paper on the profiles of competence development in upper secondary education. The authors expected to identify generalised reading and mathematical competence profiles and unique profiles where one develops faster than the other. As a result, the profiles could then be predicted by students' interests and vocational education.

In this study, the results show only one relevant profile. The findings draw on the significant limitations of the study due to the results and included the omission of hypothesis two and three. In the conclusion section, the authors suggested that the competence profiles are questionable due to limitations in the analysis given the context of the study. This is an important finding to see if conditions hold true in other contexts. The overall methodology used to conduct the study is robust and the reviewer has no concerns present on the validity of the results.

This reviewer wonders if the single profile identified can be explained more in the discussion? Furthermore, what can the authors say about the results contradicting or corroborating with existing literature? What kind of recommendations for future research in this area can the authors give from the findings in this paper, such as avoiding the pitfalls and any other approaches that are needed to be explored?

It is advised that colloquial terms in the manuscript are minimised where possible, such as could, would, should, may, etc. For example, line 127-128 contains “should therefore also more likely show”. This is but one example that requires a change to shorter and more assertive statements. Doing this will enrich the quality of the paper substantially.

Reviewer #3: I have carefully reviewed this manuscript (ms.). I must congratulate the authors of this ms. for several reasons:

This work is robustly justified taking into account the previous empirical evidence found in Germany. Therefore, the objectives and the established hypotheses are consistent with the recruited sample in Germany.

Along these lines, this reviewer wishes to congratulate the authors of this ms. since this fact is not common. Thus, this reviewer, after reviewing more than 150 ms. per year (many included in Q1, Q2 of the WoS-JCR) do not find articles as well organized (culturally focused), structured, written and robustly justified as this ms. Sincerely, congratulations!

Furthermore, the type of sampling, the size of the sample and the statistical analyzes conducted are fine and up-to-date. It's GREAT!

In any case, I would like to make some suggestions for the improvement of this ms., which can be published in PLOS ONE, of course:

TITLE:

The title of this ms. it should end by using the expression ".... in a sample of Germany students". This helps to quickly focus the reader of this ms.

ABSTRACT

Indicate type of sampling conducted.

Indicate the name of the administered instruments.

Specify type of statistical analysis conducted in this study.

INTRODUCTION

This section should include more information regarding the German educational system. Authors should bear in mind that PLOS ONE is an international journal and, therefore, authors should be more precise when describing what the German educational system is like and what it consists of. This will undoubtedly make it even easier to understand this ms. by PLOS ONE readers.

DISCUSSION

Authors should robustly accept or reject the hypotheses initially raised in the Introduction section.

Furthermore, it is essential that the authors present the main practical implications of the results found in this ms. both for researchers and applied professionals (mainly, school psychologists and other applied professionals).

7. PLOS authors have the option to publish the peer review history of their article (what does this mean?). If published, this will include your full peer review and any attached files.

Reviewer #1: **Yes: **Carmen Álvarez Álvarez

Reviewer #2: No

Reviewer #3: **Yes: **Candido J. Ingles

---

## [Author Response · Author response to Decision Letter 0]

23 Aug 2021

We want to sincerely thank the reviewers for their work in giving advise and suggestions to further improve the paper. We hope, that all reviewers can see the advances made by the authors to improve the paper. 

A full reviewer and editor response can be found in the document "response to reviewers"

---

## [Decision Letter · Decision Letter 1]

20 Sep 2021

Determinants of profiles of competence development in mathematics and reading in upper secondary education in Germany

PONE-D-21-17237R1

Dear Dr. Freund,

We’re pleased to inform you that your manuscript has been judged scientifically suitable for publication and will be formally accepted for publication once it meets all outstanding technical requirements.

Kind regards,

Vitomir Kovanovic, Ph.D.

Academic Editor

PLOS ONE

Additional Editor Comments (optional):

Reviewers' comments:

Reviewer's Responses to Questions

**Comments to the Author**

1. Does the manuscript adhere to the experimental procedures and analyses described in the Registered Report Protocol?

If the manuscript reports any deviations from the planned experimental procedures and analyses, those must be reasonable and adequately justified.

Reviewer #2: Yes

Reviewer #3: Yes

2. If the manuscript reports exploratory analyses or experimental procedures not outlined in the original Registered Report Protocol, are these reasonable, justified and methodologically sound?

A Registered Report may include valid exploratory analyses not previously outlined in the Registered Report Protocol, as long as they are described as such.

Reviewer #2: Yes

Reviewer #3: Yes

3. Are the conclusions supported by the data and do they address the research question presented in the Registered Report Protocol?

The manuscript must describe a technically sound piece of scientific research with data that supports the conclusions. The conclusions must be drawn appropriately based on the research question(s) outlined in the Registered Report Protocol and on the data presented.

Reviewer #2: Yes

Reviewer #3: Yes

4. Have the authors made all data underlying the findings in their manuscript fully available?

Reviewer #2: Yes

Reviewer #3: Yes

5. Is the manuscript presented in an intelligible fashion and written in standard English?

Reviewer #2: Yes

Reviewer #3: Yes

6. Review Comments to the Author

Please use the space provided to explain your answers to the questions above. (Please upload your review as an attachment if it exceeds 20,000 characters)

Reviewer #2: Thank you for addressing the relative feedback and expanding on the discussion and implications. The paper reads well and the findings discussed, included in the implications, now provide clearer value and contribution.

Reviewer #3: The authors have made a great effort to improve the scientific quality of this manuscript (ms.).

The revised version of this ms. it should be accepted for publication in PLOS ONE.

7. PLOS authors have the option to publish the peer review history of their article (what does this mean?). If published, this will include your full peer review and any attached files.

Reviewer #2: No

Reviewer #3: **Yes: **Candido J. Ingles

---

## [Editor Report · Acceptance letter]

24 Sep 2021

PONE-D-21-17237R1 

Determinants of profiles of competence development in mathematics and reading in upper secondary education in Germany 

Dear Dr. Freund:

I'm pleased to inform you that your manuscript has been deemed suitable for publication in PLOS ONE. Congratulations! Your manuscript is now with our production department. 

Kind regards, 

on behalf of

Dr. Vitomir Kovanovic 

Academic Editor

PLOS ONE